# Assessment of blood pressure measurement technique amongst nurses working at a tertiary care cardiac center

Muhammad Younis[1], Khalid Iqbal Bhatti📧[1]*, Kalsoom Chachar[1], Paras Nazir[1], Javaria Rafique[1], Areesha Khalid[2], Sanjana Karera[1], Fawad Farooq[1], Abdul Hakeem[1], Tahir Saghir[1], Jawaid Akbar Sial[1]

1 National Institute of Cardiovascular Diseases (NICVD), Karachi, Pakistan, 2 Baqai Medical University, Karachi, Pakistan

* khalidsaindad@yahoo.com

**Data Availability Statement:** All relevant data are within the paper and its Supporting Information files.

## Abstract

### Background

Accurate measurement is indispensable for effectively managing hypertension (HTN); any error in technique or instrumentation can lead to misdiagnosis and improper management. Thus, the present study aimed to assess the knowledge and skills of blood pressure (BP) measurement among nurses at a tertiary care cardiac center in Karachi.

### Materials and methods

Nursing staff responsible for BP assessment at various stations were identified, observed, and interviewed to evaluate their skill and knowledge levels regarding BP measurement techniques. Nurses' skill levels were assessed using a checklist based on the American Heart Association (AHA) guidelines for BP assessment.

### Results

Seventy-five nurses participated in the study, with 49 (65.3%) being male and a mean age of 32.1 ± 6.2 years. Only 25 (33.3%) nurses reported reading the AHA guidelines for BP measurement. None of the nurses demonstrated excellent skills; 19 (25.3%) showed good skills, while 56 (74.7%) showed poor skills in BP measurement. A poor compliance was observed on a total of 14/31 steps with compliance rate of less than 50%. Similarly, none of the nurses exhibited excellent knowledge; only 3 (4%) had good knowledge, while 72 (96%) had poor knowledge about BP measurement. A poor knowledge was observed on a total of 18/36 items with correct response rate of less than 50%.

### Conclusion

Nurses working at various stations of a tertiary cardiac center exhibited inadequate skills and knowledge regarding BP measurement. This underscores the necessity for comprehensive training and education to enhance the accurate assessment of BP.

**Funding:** The author(s) received no specific funding for this work.

**Competing interests:** The authors have declared that no competing interests exist.

## Introduction

Hypertension, defined as the persistent elevation of systemic arterial pressure above a predetermined threshold, is a widespread medical condition affecting approximately 1 billion individuals globally as of current estimates [1]. The World Health Organization (WHO) anticipates this figure to escalate to 1.56 billion by 2025, encompassing nearly 29.2% of the global populace [1]. Over the past few decades, the prevalence of hypertension among adults worldwide has surged from 594 million in 1975 to 1.13 billion in 2015 [2]. In developed nations, 30% of adults are hypertensive, a figure projected to soar to 60% in the forthcoming years [3]. In Pakistan, epidemiological studies have reported varying rates of hypertension, with figures ranging from 14% to 33% among adults [4–6].

Hypertension serves as a modifiable risk factor for numerous severe health conditions, including peripheral vascular disease, end-stage renal disease, stroke, myocardial infarction, and congestive heart failure, with it being the leading cause of mortality worldwide [7]. Uncontrolled hypertension significantly elevates the risk of heart attack, heart failure, kidney diseases, stroke, lifelong disability, and death [8]. While pharmacological interventions effectively mitigate the risk of coronary heart disease and other cardiovascular complications, non-pharmacological approaches such as lifestyle modifications also play a pivotal role in reducing morbidity and mortality among hypertensive patients [9].

Early detection of hypertension through screening and prompt initiation of treatment prior to the onset of target organ damage can prevent a substantial number of fatalities [10]. Despite clear guidelines on blood pressure measurement techniques, significant inter-observer variations persist, attributed to observer bias, faulty equipment, and lack of standardization in measurement techniques [11]. Studies have indicated suboptimal levels of knowledge, training, and awareness among healthcare professionals regarding blood pressure measurement techniques and equipment [12–15]. Accurate measurement is imperative for proper hypertension management, as any technical or instrumental error can lead to inaccurate diagnosis and subsequent mismanagement [15, 16].

Given that blood pressure measurement is frequently conducted by nurses in clinical settings, ensuring their proficiency and knowledge in this regard is crucial. However, local data on this aspect are often lacking. Therefore, this study aims to assess the knowledge and skills of blood pressure measurement among nurses at the largest tertiary care cardiac center in Karachi, Pakistan, with the goal of identifying any knowledge gaps and facilitating skill development to enhance hypertension management.

## Methods

This cross-sectional study, a Knowledge, Attitude, and Practice (KAP) survey, was conducted at the National Institute of Cardiovascular Diseases (NICVD) in Karachi, Pakistan, the largest cardiac care center in the country. The research was undertaken as part of a fellowship in cardiology from the College of Physicians and Surgeons Pakistan (CPSP). Ethical approval was obtained from the institutional review board (IRB), and all participating nurses were briefed about the study's purpose and potential benefits. Verbal informed consent was obtained from each nurse regarding their participation and the publication of study results. Due to observational nature of the study, IRB waived the requirement of written consent and approved the use of verbal consent. The study spanned a period of six months, from 01/12/2021 to 01/06/2022.

Following approval, nursing staff meeting the inclusion criteria were observed and interviewed to evaluate their skill and knowledge levels regarding blood pressure (BP) measurement techniques. Nurses responsible for BP assessment across various stations, including the

emergency department (ER), outpatient department (OPD), and hospital floors, were identified using a roster of nursing staff. The study employed a non-probability consecutive sampling technique. Exclusion criteria comprised doctors or non-nursing healthcare providers, nurses with less than six months of experience, and those who declined consent.

Observations of nurses' skills were conducted in real-time as they assessed BP in patients, using a checklist based on the American Heart Association (AHA) guidelines [17]. The same sphygmomanometer was used for all nurses, provided by the principal investigator. Skill levels were categorized as excellent ($\geq$80% steps correctly followed), good (60% to 79% steps correctly followed), or poor (<60% steps correctly followed).

Following the observations, nurses completed a self-administered questionnaire to assess their knowledge of BP measurement. Knowledge scores were calculated based on the percentage of correct responses, with categories defined as excellent ($\geq$80% correct answers), good (60% to 79% correct answers), or poor (<60% correct answers).

Demographic data, including gender, age, qualification, years of experience, and time since last BP assessment training, were recorded on a structured proforma. To mitigate observer bias, a single observer with a predefined checklist was dedicated to all participants.

Sample size for the study was calculated with an expected percentage of poor knowledge regarding BP measurement as 19.6% amongst nurses [11], 95% confidence level and 9% margin of error the minimum required sample size for this study was calculated to be 75. Data were analyzed using IBM SPSS version 21, with summary statistics computed for continuous variables and frequency distributions for categorical variables. Confounding variables were controlled through stratification, and post-stratification tests, such as the Chi-square test or Fisher's Exact test, were applied to assess the impact of confounders on skill and knowledge levels. A p-value $\leq$ 0.05 was considered statistically significant.

## Results

A total of 75 nurses were observed and interviewed in this study, out of which 49 (65.3%) were male and mean age was 32.1 ± 6.2 years. Only 25 (33.3%) read the AHA guidelines for BP measurement. None of the nurse showed excellent skills, 19 (25.3%) nurses showed good and 56 (74.7%) showed poor BP measurements skills. None of the nurse showed excellent knowledge, 3 (4%) nurses showed good and 72 (96%) showed poor knowledge about BP measurements (Table 1).

The distribution of PB measurement skill and knowledge level showed no significant association with any of the baseline characteristics, except for the nurses who read AHA guidelines for BP measurement, they showed relatively good knowledge (p = 0.012), Table 2.

The distribution of steps correctly followed by nurses using a 31-item checklist based on the AHA guidelines for BP assessment is presented in Fig 1. A poor compliance was observed on a total of 14/31 steps with compliance rate of less than 50%.

The distribution of correct response by nurses to the 36-item questionnaire knowledge assessment based on the AHA guidelines for BP assessment is presented in Fig 2. A poor knowledge was observed on a total of 18/36 items with correct response rate of less than 50%.

## Discussion

The findings of this study underscore the critical importance of ensuring accurate BP measurement, particularly in the context of managing hypertension. The results reveal a concerning gap in both the skills and knowledge of BP measurement among nurses at a tertiary care cardiac center in Karachi. With hypertension being a leading cause of morbidity and mortality globally, the implications of such deficiencies in healthcare delivery are significant.

**Table 1. Distribution of baseline characteristic, blood pressure measurement skill level, and knowledge level.**

| Characteristics | Summary |
|---|---|
| **Total (N)** | **75** |
| **Gender** | |
| Male | 49 (65.3%) |
| Female | 26 (34.7%) |
| **Age (years)** | 32.1 ± 6.2 |
| **Qualification** | |
| Degree | 61 (81.3%) |
| Diploma | 13 (17.3%) |
| Certificate | 1 (1.3%) |
| **Years of experience** | 9.7 ± 5.6 |
| Up to 5 years | 18 (24%) |
| 6 to 10 years | 34 (45.3%) |
| > 10 years | 23 (30.7%) |
| **Read the AHA guidelines for blood pressure measurement** | |
| No | 50 (66.7%) |
| Yes | 25 (33.3%) |
| **Years since last BP assessment training** | 9.8 ± 5.8 |
| 1 to 2 years | 2 (2.7%) |
| 3 to 4 years | 3 (4%) |
| > 4 years | 70 (93.3%) |
| **Skills Level** | 54.5 ± 9.1 |
| Poor Skills | 56 (74.7%) |
| Good Skills | 19 (25.3%) |
| Excellent Skills | 0 (0%) |
| **Knowledge level** | 48.2 ± 7.9 |
| Poor Knowledge | 72 (96%) |
| Good Knowledge | 3 (4%) |
| Excellent Knowledge | 0 (0%) |

The fact that only a third of the nurses reported reading the AHA guidelines for BP measurement highlights a potential lack of awareness or emphasis on best practices in this crucial aspect of patient care. Additionally, mean years since the last BP measurement training was 9.8 ± 5.8 years with 93.3% of the nurses with last BP measurement training more than 4 years ago. This finding suggests a need for greater attention to training and educational initiatives aimed at familiarizing healthcare professionals with standardized guidelines and protocols for BP assessment.

The observed poor compliance rates and suboptimal skill levels among the majority of nurses further emphasize the need for targeted interventions to improve proficiency in BP measurement techniques. Inaccurate or inconsistent measurement practices can result in misdiagnosis, improper management, and ultimately, compromised patient outcomes [18]. Several studies have examined BP measurement practices among healthcare workers, revealing varying levels of adherence to guidelines and recommendations. Machado et al. [19] reported that only 65% of BP measurement steps were followed by nurses. In contrast, a study involving physicians indicated good adherence to BP measurement guidelines, although adherence evaluation was self-reported [20]. In this study poor compliance was observed on a total of 14/31 steps with compliance rate of less than 50%. These steps included; checking the stopcock on the bulb for any malfunction, advising patient to stay quite during BP taking, inflating the

**Table 2. Distribution of blood pressure measurement skill level and knowledge level by baseline characteristic.**

| | Skills Level | | | Knowledge level | | |
|---|---|---|---|---|---|---|
| | **Poor** | **Good** | **Excellent** | **Poor** | **Good** | **Excellent** |
| **Gender** | | | | | | |
| Male | 34 (69.4%) | 15 (30.6%) | 0 (0%) | 47 (95.9%) | 2 (4.1%) | 0 (0%) |
| Female | 22 (84.6%) | 4 (15.4%) | 0 (0%) | 25 (96.2%) | 1 (3.8%) | 0 (0%) |
| P-value | 0.149 | | | 0.960 | | |
| **Qualification** | | | | | | |
| Degree | 46 (75.4%) | 15 (24.6%) | 0 (0%) | 58 (95.1%) | 3 (4.9%) | 0 (0%) |
| Diploma | 9 (69.2%) | 4 (30.8%) | 0 (0%) | 13 (100%) | 0 (0%) | 0 (0%) |
| Certificate | 1 (100%) | 0 (0%) | 0 (0%) | 1 (100%) | 0 (0%) | 0 (0%) |
| P-value | 0.756 | | | 0.699 | | |
| **Years of experience** | | | | | | |
| Up to 5 years | 14 (77.8%) | 4 (22.2%) | 0 (0%) | 18 (100%) | 0 (0%) | 0 (0%) |
| 6 to 10 years | 25 (73.5%) | 9 (26.5%) | 0 (0%) | 32 (94.1%) | 2 (5.9%) | 0 (0%) |
| > 10 years | 17 (73.9%) | 6 (26.1%) | 0 (0%) | 22 (95.7%) | 1 (4.3%) | 0 (0%) |
| P-value | 0.941 | | | 0.585 | | |
| **Read the AHA guidelines for blood pressure measurement** | | | | | | |
| No | 37 (74%) | 13 (26%) | 0 (0%) | 50 (100%) | 0 (0%) | 0 (0%) |
| Yes | 19 (76%) | 6 (24%) | 0 (0%) | 22 (88%) | 3 (12%) | 0 (0%) |
| P-value | 0.851 | | | 0.012 | | |
| **Years since last BP assessment training** | | | | | | |
| 1 to 2 years | 1 (50%) | 1 (50%) | 0 (0%) | 2 (100%) | 0 (0%) | 0 (0%) |
| 3 to 4 years | 2 (66.7%) | 1 (33.3%) | 0 (0%) | 3 (100%) | 0 (0%) | 0 (0%) |
| > 4 years | 53 (75.7%) | 17 (24.3%) | 0 (0%) | 67 (95.7%) | 3 (4.3%) | 0 (0%) |
| P-value | 0.675 | | | 0.894 | | |

pressure 20 mmHg above radial pulse disappearance before started listening, waiting for 1 minute before taking 2nd reading, ensuring patient has emptied his/her bladder, asking questions about recent smoking, exercise, coffee/tea intake before BP taking, recording BP via palpatory method, palpating radial artery for palpatory method, measuring BP twice on a visit, using both arms for BP measurement on first visit, checking orthostatic hypotension, checking pulse for orthostatic hypotension, and waiting for 2–3 minutes after taking BP in lying position for orthostatic hypotension. Therefore, efforts to address these deficiencies should include not only education on proper technique but also ongoing monitoring and reinforcement of skills through regular training and quality assurance initiatives.

Similarly, the widespread lack of knowledge regarding BP measurement, with the vast majority of nurses demonstrating poor understanding of key concepts, is concerning. Several studies, including those by Machado et al. [21] and Machado et al. [19], have highlighted a pervasive inadequacy in knowledge regarding BP measurement among healthcare workers. Machado et al. [21] found moderate levels of BP measurement knowledge across various domains, while another study by Machado et al. [19] specifically noted poor theoretical knowledge among cardiac nurses in indirect BP measurement. Additionally, a study conducted in Nigeria revealed that the majority of healthcare workers lacked familiarity with guidelines and exhibited a low level of BP measurement knowledge [22]. This suggests a need for comprehensive educational programs that not only cover the technical aspects of measurement but also ensure a thorough understanding of the underlying principles and factors that may influence blood pressure readings.

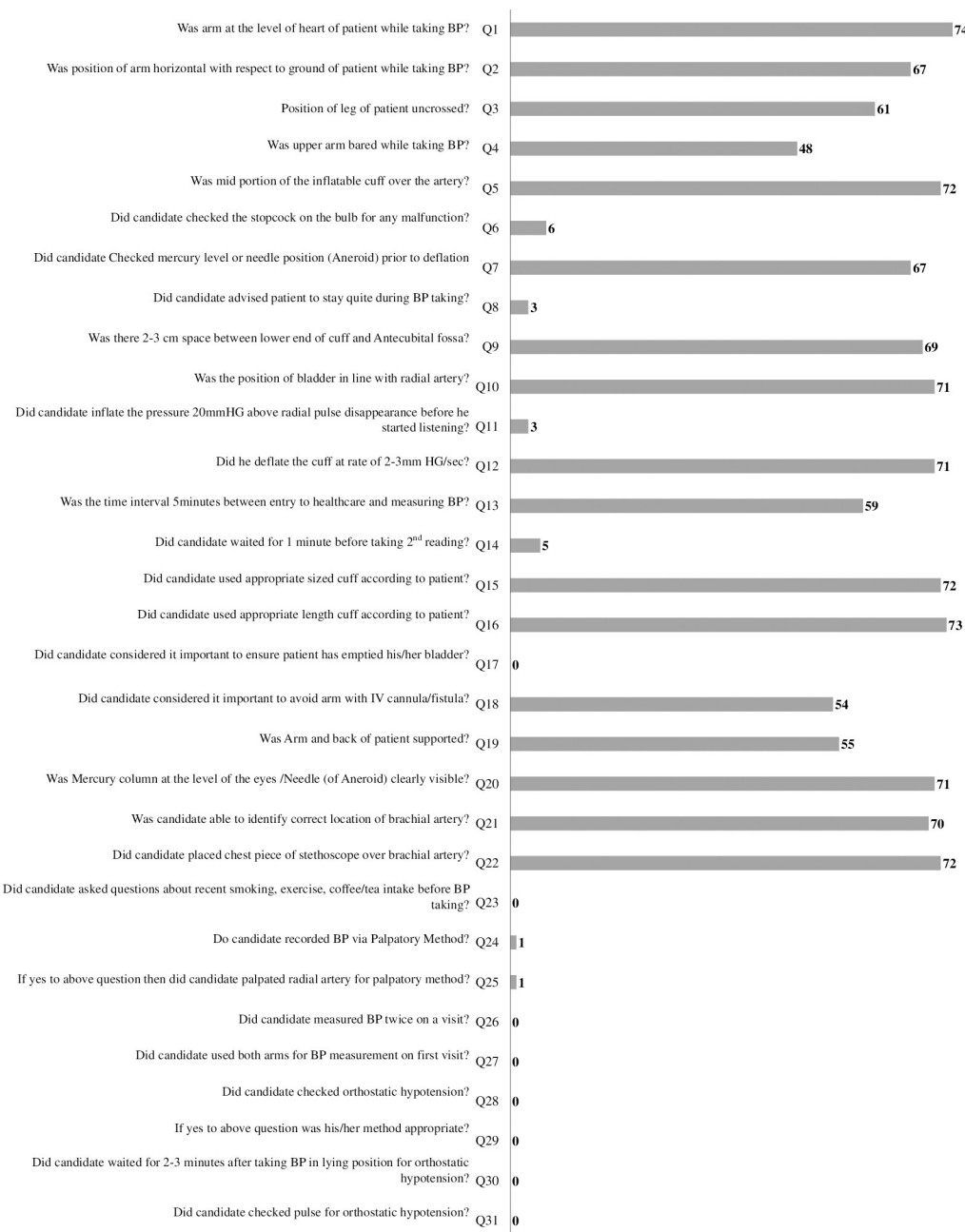

**Fig 1. Distribution of steps correctly followed by nurses using a 31-item checklist based on the American Heart Association (AHA) guidelines for BP assessment.**

Elzeky ME et al. [23] assessed the effectiveness of using WhatsApp as a distance education tool to improve BP measurement knowledge and accuracy among nurses. Results showed that after the intervention, the WhatsApp group demonstrated significantly higher knowledge scores and reduced errors in BP measurement compared to the control group. However, there was no significant difference in performance scores between the two groups, suggesting that a multimodal approach may be needed to enhance practical skills in addition to knowledge. Block et al. [24], who noted that a web-based educational program improved knowledge but

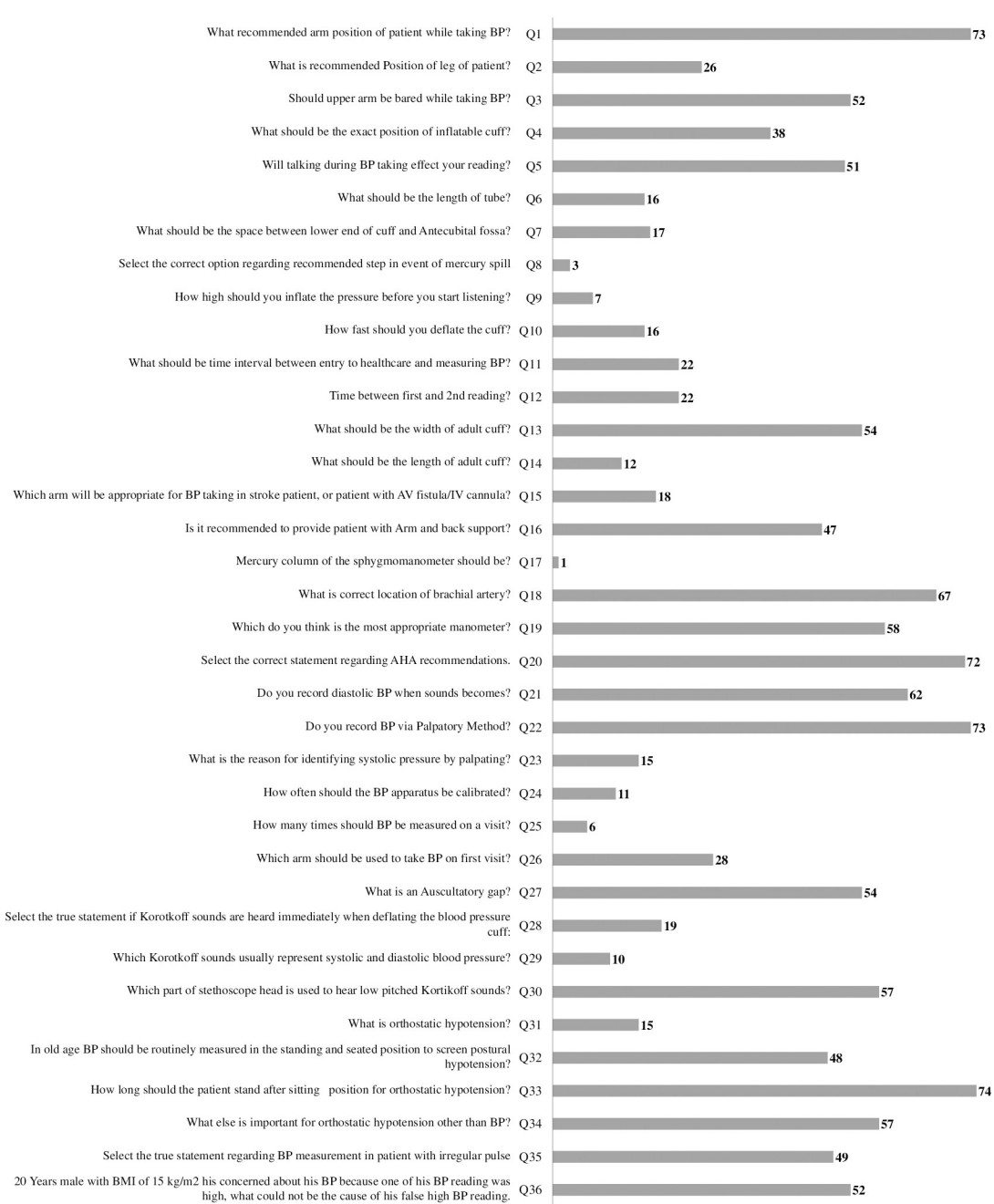

**Fig 2. Distribution of correct response by nurses to the 36-item questionnaire knowledge assessment based on the American Heart Association (AHA) guidelines for BP assessment.**

not skills or attitude. Block et al. suggested a mix of live and web-based training may be needed for attitude and skill improvement. Conversely, Rabbia et al. [25] reported significant improvement in BP measurement technique after a 1-day face-to-face training program, and Machado et al. [21] found similar results with a 2-hour face-to-face training. Additionally, a study implementing a bundle intervention program saw significant improvement in BP measurement performance scores, attributed to addressing barriers and enhancing organizational awareness [26].

While our study comprehensively addressed both knowledge and skill levels through direct observation, it is important to acknowledge certain limitations. Primarily, our reliance on a single center and a relatively small sample size may restrict the generalizability of our findings. To ensure more robust and comprehensive conclusions, future research endeavors should aim for larger sample sizes encompassing diverse levels of medical and nursing education. This broader inclusion would provide a more accurate representation of BP measurement proficiency across various healthcare settings and educational backgrounds.

## Conclusion

Nurses working at various stations of a tertiary cardiac center exhibited inadequate skills and knowledge regarding BP measurement. This underscores the importance of prioritizing training and education initiatives to enhance the skills and knowledge of healthcare professionals, particularly nurses, in BP measurement. By addressing these deficiencies, healthcare organizations can improve the quality of care provided to patients with hypertension, ultimately reducing the risk of complications and improving long-term outcomes.

## Supporting information

**S1 Dataset. Dataset in excel format.**
(XLSX)

## Acknowledgments

The authors wish to acknowledge the support of the Clinical Research Department staff members of the National Institute of Cardiovascular Diseases (NICVD), Karachi, Pakistan.

## Author Contributions

**Conceptualization:** Khalid Iqbal Bhatti.

**Data curation:** Khalid Iqbal Bhatti.

**Formal analysis:** Sanjana Karera.

**Investigation:** Khalid Iqbal Bhatti.

**Methodology:** Javaria Rafique.

**Project administration:** Muhammad Younis, Kalsoom Chachar.

**Supervision:** Khalid Iqbal Bhatti, Paras Nazir.

**Validation:** Fawad Farooq.

**Visualization:** Paras Nazir, Fawad Farooq.

**Writing – original draft:** Areesha Khalid.

**Writing – review & editing:** Abdul Hakeem, Tahir Saghir, Jawaid Akbar Sial.

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
