## [Decision Letter · Decision Letter 0]

25 Jul 2024

Assessment of Blood Pressure Measurement Technique amongst Nurses Working at a Tertiary Care Cardiac Center

PONE-D-24-06159

Dear Dr. Bhatti,

We’re pleased to inform you that your manuscript has been judged scientifically suitable for publication and will be formally accepted for publication once it meets all outstanding technical requirements.

Kind regards,

Fahad Jibran Siyal, Ph.D.

Academic Editor

PLOS ONE

2. In the ethics statement in the Methods, you have specified that verbal consent was obtained. Please provide additional details regarding how this consent was documented and witnessed, and state whether this was approved by the IRB.

4. PLOS requires an ORCID iD for the corresponding author in Editorial Manager on papers submitted after December 6th, 2016. Please ensure that you have an ORCID iD and that it is validated in Editorial Manager. To do this, go to ‘Update my Information’ (in the upper left-hand corner of the main menu), and click on the Fetch/Validate link next to the ORCID field. This will take you to the ORCID site and allow you to create a new iD or authenticate a pre-existing iD in Editorial Manager. Please see the following video for instructions on linking an ORCID iD to your Editorial Manager account: https://www.youtube.com/watch?v=_xcclfuvtxQ.

Comments from PLOS Editorial Office: We note that one or more reviewers has recommended that you cite specific previously published works. As always, we recommend that you please review and evaluate the requested works to determine whether they are relevant and should be cited. It is not a requirement to cite these works. We appreciate your attention to this request.

Reviewers' comments:

Reviewer's Responses to Questions

**Comments to the Author**

1. Is the manuscript technically sound, and do the data support the conclusions?

Reviewer #1: Yes

Reviewer #2: Yes

2. Has the statistical analysis been performed appropriately and rigorously? 

Reviewer #1: Yes

Reviewer #2: Yes

3. Have the authors made all data underlying the findings in their manuscript fully available?

Reviewer #1: Yes

Reviewer #2: Yes

4. Is the manuscript presented in an intelligible fashion and written in standard English?

Reviewer #1: Yes

Reviewer #2: Yes

5. Review Comments to the Author

Reviewer #1: This paper is interesting and novel and addresses an important gap in the current literature.

It si overall well written, but the discussion that should better highlight the importance of the vascular nurse. For this porpoise cite and comment the following paper: Ielapi, Nicola et al. “Vascular Nursing and Vascular Surgery.” Annals of vascular surgery vol. 68 (2020): 522-526. doi:10.1016/j.avsg.2020.05.038.

Reviewer #2: This article provides an insightful and thorough assessment of blood pressure measurement techniques amongst nurses, highlighting critical areas for improvement and training. The meticulous methodology and detailed analysis enhance the study's credibility and relevance.

6. PLOS authors have the option to publish the peer review history of their article (what does this mean?). If published, this will include your full peer review and any attached files.

Reviewer #1: No

Reviewer #2: **Yes: **Dr. Mir Hassan Khoso PhD, Associate Professor of Biochemistry

---

## [Editor Report · Acceptance letter]

1 Aug 2024

PONE-D-24-06159 

PLOS ONE

Dear Dr. Bhatti, 

I'm pleased to inform you that your manuscript has been deemed suitable for publication in PLOS ONE. Congratulations! Your manuscript is now being handed over to our production team.

Kind regards, 

on behalf of

Dr. Fahad Jibran 

Academic Editor

PLOS ONE